# Phenolic Profiles, Antioxidant, and Hypoglycemic Activities of *Ribes meyeri* Fruits

**DOI:** 10.3390/foods12122406

**Published:** 2023-06-18

**Authors:** Le Zhang, Qiang Wang, Yayun Zhao, Juan Ge, Dajun He

**Affiliations:** 1College of Life Science, Shihezi University, Shihezi 832003, China; zhangle@stu.shzu.edu.cn (L.Z.); wangqiang1@stu.shzu.edu.cn (Q.W.); z15729932636@163.com (Y.Z.); 2Analysis and Testing Center, Shihezi University, Shihezi 832003, China

**Keywords:** *Ribes meyeri*, phenolics, anthocyanins, antioxidant activity, antidiabetic activity

## Abstract

*Ribes meyeri* is a *Ribes* genus in the Saxifragaceae family, which is used as both medicine and food. However, the active components and biological activities of *R. meyeri* fruits are still unknown. In this paper, the phenolic components and their antioxidant and hypoglycemic activities of *R. meyeri* fruits were studied. Firstly, a total of 42 phenolic components of *R. meyeri* fruits, including 26 anthocyanins, 9 flavonoids, and 7 phenolic acids, were tentatively identified using HPLC-QTOF-MS/MS, and the main four anthocyanins were quantified using UPLC-MS/MS. The result indicated that cyanidin-3-*O*-rutinoside is the main anthocyanin in the *R. meyeri* fruits. The anthocyanin fraction of *R. meyeri* fruits exhibited significant inhibitory activity on *α*-amylase and *α*-glucosidase. The anthocyanin fraction from *R. meyeri* fruits significantly increased the glucose uptake of 3T3-L1 adipocytes. This is the first study of a qualitative and quantitative analysis of the phenolics of *R. meyeri* fruits.

## 1. Introduction

In the human diet, berries are the most commonly consumed fruit. There is growing evidence that the edible and colorful fruits have various health-promoting and disease-preventing benefits, which are related to a variety of bioactive phytochemicals, including phenolic compounds [1,2].

Black currants and red currants are favorite *Ribes* berries. Researchers have found that currants contain a lot of polyphenols, particularly anthocyanins. The main anthocyanins of black currants are glucosides and rutinosides of cyanidin and delphinidin [3]; those of red currants are the cyanidin-diglycosides [4], including phenolic acid derivatives, flavonols, as well as proanthocyanidins [5]. Increasing evidence suggests that black currant fruit has a variety of bioactive activity, such as antioxidant, anti-inflammatory, anti-tumor, antidiabetic, and so on [6,7,8,9].

Unlike the mainstream berries on the market, which might contain residues of pesticides and chemicals, wild plants require no irrigation water or fertilizers, and their wild berries are considered a healthy alternative [10,11]. *Ribes meyeri* fruit is a unique, small, black edible berry that is widely distributed in Xinjiang, China, as shown in Figure 1, and is used in Uyghur medicine to reduce blood pressure and blood lipids and to treat menstrual disorders [12]. In our previous study, we found that *Ribes meyeri* fruit is a rich source of phenolics, specifically anthocyanin [13], and shows a good potential to become an alternative for commercial *Ribes* fruits such as *Ribes nigrum*. However, research on the chemical constituents and biological activities of *Ribes meyeri* fruits is scarce. In order to make better use of *Ribes meyeri* fruits, it is essential to establish a rapid and accurate method to discover the active ingredients.

α-Glucosidase is an important target for screening hypoglycemic activity that catalyzes the final stage of carbohydrate digestion. α-Glucosidase inhibitors can slow glucose release from complex carbohydrates and delay glucose absorption, resulting in lower postprandial plasma glucose levels [14]. α-Amylase is one of the key enzymes involved in the carbohydrate digestion process by catalyzing the hydrolysis of the α-D-1,4-glucosidic bond in starch [6]. Therefore, the inhibition of α-amylase activity is considered an effective strategy to regulate blood sugar levels and manage food-linked hyperglycemia [15,16]. In addition to the two conventional screening targets for hypoglycemic activity mentioned above, protein tyrosine phosphatase 1B (PTP1B) is currently one of the most promising targets for the treatment of type 2 diabetes (T2D). PTP1B is a negative modulator of insulin signaling and plays an important role in insulin resistance. However, most PTP1B inhibitors failed in clinical trials because of poor clinical efficacy and severe side effects. Therefore, it is essential to discover novel PTP1B inhibitors for the development of new drugs [17,18]. In this study, we screened the hypoglycemic activity of *Ribes meyeri* fruits using these three important targets.

The combination of high-performance liquid chromatography with quadrupole time-of-flight mass spectrometry (HPLC-QTOF/MS) has been recognized as a highly sensitive and accurate analytical tool to identify the phenolics of medicinal and edible plants [19]. In this study, the phenolics of *R. meyeri* fruits were characterized using HPLC-QTOF/MS; meanwhile, their antioxidative and antidiabetic activities were determined at the molecular and cellular levels. To date, this is the first comprehensive study to analyze the phenolics and evaluate the antioxidative and hypoglycemic properties of *R. meyeri* berries. It will provide a reference for enriching edible *Ribes* resources.

## 2. Materials and Methods

### 2.1. Plant Material and Reagents

In August 2021, *Ribes meyeri* fruits were collected in Manasi County South Mountain Forestry, Xinjiang, China, and identified by Associate Prof. Wenbin XU. Then they were stored at −20 °C. The *Ribes meyeri* fruit specimen (RML210811) was coded and maintained in the botany laboratory of Shihezi University.

Four standards of anthocyanin, cyanidin-3-*O*-rutinoside, delphinidin-3-*O*-rutinoside, cyanidin-3-*O*-glucoside, and delphinidin-3-*O*-glucoside, were purchased from Chengdu Durst Biotechnology Co. (Chengdu, China). Formic acid, acetonitrile, and methanol (Optima™ LC/MS Grade) were purchased from Fisher Scientific (Waltham, MA, USA). The following analytical purity chemicals were provided by Adamas Co. (Shanghai, China): methanol, hydrochloric acid, potassium chloride, ethanol, potassium persulfate, and sodium potassium tartrate. A macroporous resin, AB-8, was obtained from Cangzhou Bao’en Adsorption Material Technology Co. (Cangzhou, China).

Protein tyrosine phosphatase 1B (PTP1B, EC 3.1.3.48) from human recombinant protein, *α*-glucosidase (EC 3.2.1.20) from *Saccharomyces cerevisiae*, *α*-amylase (EC 3.2.1.1) from porcine pancreas, *p*-nitrophenyl-β-D-glucoside (*p*NPG), Na_3_VO_4_, ascorbic acid, acarbose, 2,2′-azino-bis (3-ethylbenzthiazoline-6-sulfonic acid) (ABTS), and 2,2-diphenyl-1-picrylhydrazyl (DPPH) were provided by Sigma-Aldrich Chemical Co. (St Louis, MO, USA).

### 2.2. Extraction Procedure

One hundred grams of *R. meyeri* fruits were homogenized with a homogenizer, added to 1000 mL methanol (including 0.1% hydrochloric acid), and then ultrasonically extracted at room temperature for 30 min. The extraction procedure was repeated three times. Then the extracts were filtered, and the filtrate was centrifuged at 10,000 rpm for 20 min. Subsequently, the solvent of the supernatant was removed using a rotary evaporator at 40 °C to obtain the extract (7.9 g). For LC-MS analysis, each sample was dissolved in methanol at the appropriate concentration and filtered with Acrodisc syringe filters (0.22 μm, Pall Corporation, GA, USA).

### 2.3. Purification of Anthocyanin from R. meyeri Fruits

In order to purify anthocyanin from *R. meyeri* fruits, 1.0 g of methanol extract was dissolved in 200 mL ultrapure water and added to a pretreated AB-8 macroporous resin column (100 × 5 cm). Firstly, the flow rate of the resin column was 5.0 mL/min, and the anthocyanins were absorbed on the AB-8 macroporous resin. Then, 500 mL ultrapure water was added to remove impurities. Finally, a 100 mL methanol solution containing 0.1% HCl was added to elute the anthocyanins, at a flow rate of 1.0 mL/min. The methanol eluent was combined and concentrated using a rotary evaporator at 40 °C to obtain the anthocyanin fraction (0.74 g), which was saved at −20 °C for quantitative analysis and activity evaluation.

### 2.4. Qualitative Analysis of Phenolics from R. meyeri Fruits

Chromatographic conditions: Shimadzu UFLC liquid system was equipped with SIL-20AC autosampler, two LC-20AD pumps, CTO-20AC column temperature box, DGU-20A degassing device, and CBM-20A communication bus module. A Prodigy Phenyl-3 column (150 mm × 4.6 mm, 5 μM, Phenomenex, Torrance, CA, USA) was used for chromatographic separation at a column temperature of 40 °C and a flow rate of 0.5 mL/min; 20 μL of sample were injected. The mobile phase consisted of 0.1% formic acid in water (A) and 0.1% formic acid/methanol (B), where the gradient elution was 95–5% A from 0 to 95 min, and 5% A isocratic from 95 to 110 min.

MS conditions: a four-stage rod time-of-flight MS TOF^®^ 4600 (Applied Biosystems/MDS Sciex, Framingham, MA, USA) with ESI interface was used. The MS analysis was recorded in both positive and negative ionization modes. The MS parameters were as follows: electrospray ionization (ESI) voltage of −4500 V, auxiliary gas of 50 psi, nebulizer gas of 35 psi, curtain gas of 25 psi, turbine gas temperature of 450 °C, decluster voltage of −80 V, and collision energy of 50 eV. The quality scan range was m/z 100-1100. The Analyst TF 1.7 software was used to obtain and process the MS data.

### 2.5. Quantitative Analysis of Anthocyanin from R. meyeri Fruits

#### 2.5.1. Determination of Anthocyanin Content

The anthocyanin content was determined according to the pH-differential method [20] with some modifications. In brief, the absorbance of a cyanidin-3-glucoside solution (potassium chloride–hydrochloric acid buffer solution, pH = 1.0 and 24.8 mM sodium acetate buffer solution, pH = 4.5) was recorded at 510 nm and 700 nm using a spectrophotometer. The content of anthocyanin was expressed as milligrams cyanidin-3-glucoside equivalents per gram of dry weight (mg C-3-G/g DW).

#### 2.5.2. UPLC-MS/MS Analysis

The four main anthocyanins of the methanol extract and anthocyanin fraction from *R. meyeri* fruits were quantitatively analyzed with the UPLC-MS/MS method [21]. A UPLC system and a triple quadrupole mass spectrometry with electrospray ionization (ESI) source (Xevo TOS, Waters Corp., Milford, MA, USA) was run with MassLynx software v.4.1. An ACQUITY UPLC BEH C_18_ column (50 mm × 5 mm, 1.7 μm, Waters, Milford, MA, USA) was used for the anthocyanin separation at 30 °C with a flow rate of 0.3 mL/min, and the injection volume was 1.0 µL. In order to optimize the simultaneous quantitative analysis method for the four major anthocyanins of *R. meyeri* berries with high resolution and narrow peaks, different UPLC conditions were compared, including different mobile phases (MeOH/H_2_O, ACN/H_2_O) and different modifiers (no modifier, 0.1%, 1%, and 5% formic acid). The optimized conditions were as follows: the mobile phase consisted of 5% formic acid (A) and acetonitrile (B), and the gradient elution was 0–1.0 min, 98% A; 1.0–3.0 min, 98–91% A; 3.0–4.0 min, 91–85% A; 4.0–6.0 min, 85–0% A; 6.0–6.2 min, 0–98% A; and 6.2–8.0 min, 98% A.

The positive ion mode with the multiple reaction monitoring (MRM) mode was used in MS, and the optimized MS parameters were as follows: source temperature of 150 °C, desolvation temperature of 450 °C, capillary voltage of 2.0 KV, nebulizer pressure of 7.0 bar, desolvation gas flow of 800 L/h, and cone gas flow of 150 L/h. The MS parameters of each component are shown in Table 1.

### 2.6. Determination of Antioxidant Activity

The antioxidant activity of the methanol extract and anthocyanin fraction from the *R. meyeri* fruits was determined with two colorimetric methods, the radical DPPH and ABTS radical scavenging methods, according to the previous report [22]. The absorbance was recorded at 510 nm with a Spectra Max MD5 microplate reader (Molecular Devices, San Jose, CA, USA). Ascorbic acid was the positive control. The IC_50_ was calculated using linear regression.

### 2.7. Antidiabetic Activity Assay

The *α*-amylase inhibitory activity was assayed according to the method in [22,23]. In brief, the sample solution (including the methanol extract and anthocyanin fraction from *R. meyeri* fruits, 10 μL dissolved in DMSO) and 0.5 U/mL of the *α*-amylase solution (10 μL) were mixed and incubated at 37 °C for 15 min. Subsequently, 500 μL of 1% starch solution was added and incubated at 37 °C for 10 min. The reaction was terminated by adding the DNS reagent (300 μL). The absorbance was determined at 540 nm with a Spectra Max MD5 microplate reader. Acarbose was the positive control.

The *α*-glycosidase inhibitory activity was determined according to the previous report [23]. Briefly, 20 μL of the sample solution, including the methanol extract and anthocyanin fraction from *R. meyeri* fruits, were combined with 0.22 U/mL *α*-glucosidase (20 μL) in 140 μL of phosphate buffer (0.1 M) and incubated at 37 °C for 20 min. Then, 10 mmol/L of *p*NPG solution (50 μL) as reaction substrate was added, followed by a 20-min incubation at 37 °C. The reaction was terminated by adding 80 μL of Na_2_CO_3_ solution (0.4 M). The absorbance of released *p*-nitrophenol at 405 nm was recorded with a microplate reader. Acarbose was the positive control.

The inhibitory activity of protein tyrosine phosphatase 1B (PTP1B) was measured using the color reaction of PTP1B hydrolyzing the phosphate group of *p*-nitrophenyl disodium phosphate (*p*-NPP) according to the reported method in [22]. In brief, 1 μL sample solution (dissolved in DMSO); 1 μL of PTP1B (0.115 mg/mL); and 96 μL of buffer containing 100 mM NaCl, 0.1% BSA, dithiothreitol (DTT, 1 mM), and 4-(2-hydroxyerhyl) piperazine-1-erhanesulfonic acid (HEPES, 50 mM, pH 7.3) were mixed and incubated at 37 °C for 10 min. Subsequently, 2 μL *p*-NPP (2 mM) was added to start the enzyme reaction at 37 °C. After 30 min, the reaction was terminated by adding 5 μL of NaOH (3 mol/L). Sodium vanadate was the positive control. The absorbance of the released *p*-nitrophenyl was measured at 405 nm. The inhibition rate was calculated as follows:Inhibition rate of PTP1B (%) = (A_blank_ − A_sample_)/A_blank_ × 100%.(1)

The effect of the *R. meyeri* fruits’ anthocyanin on the glucose consumption of 3T3-L1 adipocytes was evaluated according to a previous method [22]. The Shanghai Institute of Biochemistry and Cell Biology provided 3T3-L1 preadipocytes, which were routinely cultured and differentiated in DMEM and the differentiation medium (provided by Gibco1 Life Technologies, Carlsbad, CA, USA). After the 3T3-L1 adipocytes were fully differentiated and the cells’ density reached 8 × 10^3^ cells/well, they continued to be cultured under starvation conditions for 5 h. Finally, the 3T3-L1 adipocytes were exposed to the blank control (0.1% DMSO), insulin (reference compound, 100 nM), and the anthocyanin fractions of 12.5 μg/mL and 25 μg/mL at 37 °C for 18 h. The content of the glucose uptake was quantified with a glucose oxidase assay kit from Applygen Co. (Beijing, China). The results were expressed as the mean ± SD (*n* = 3).

### 2.8. Statistical Analysis

The data were presented as the mean ± SD (*n* = 3). To analyze the correlation between the anthocyanin content and the antioxidant hypoglycemic activities, the Pearson’s correlation coefficient was calculated with the statistical software R version 3.5.3, and results with *p* < 0.05 were considered to be significant.

## 3. Results

### 3.1. Qualitative Identification of the Chemical Composition of R. meyeri Fruits

The phenolics of the methanol extract from the *R. meyeri* fruits were characterized with HPLC-QTOF-MS/MS. A strategy was proposed (Appendix A) for identification of the chemical composition of *R. meyeri* fruits. Firstly, the methanol extract of *R. meyeri* fruits was analyzed in both positive and negative ion modes. Then, the important information, including accurate mass, the matched molecular formula, the pseudo-molecular ions, and the MS/MS product ions, was acquired. Finally, the chemical composition of *R. meyeri* fruits was qualitatively identified based on the typical product ions, fragment ions search, and the typical neutral losses. With this strategy, 42 chemical compositions were tentatively characterized from the methanol extract from *R. meyeri* fruits. In the positive ion mode, 26 anthocyanins were tentatively characterized using TIC chromatograms as shown in Figure 2. In the negative ion mode, nine flavonoids and seven phenolic acids were tentatively characterized. The TIC chromatogram is shown in Figure 2. The molecular formula, error, main product ions, and retention time of the flavonoids, phenolic acids, and anthocyanins are shown in Table 2 and Table 3.

**Identification of anthocyanins**. Compound **42** shows its molecular ion at m/z 287.0563. The molecular formula is C_15_H_11_O_6_, calculated using accurate mass data. Compound **42** was identified as cyanidin by comparison of the retention time, precursor ions, high-resolution MS data, and fragmentation ions’ MS/MS data with those of the reference standard. The typical fragmentation ions of m/z 241 and m/z 213 show in its MS/MS spectra. Compound **22** shows its precursor ion at m/z 449.1097; its intense fragmentation ion of m/z 287 [M-C_6_H_10_O_5_]^+^ indicates the loss of a hexosyl (162 Da). The typical MS/MS fragmentation ions of m/z 287, 241, and 213 suggest the existence of a cyanidin moiety. Then, compound **22** was deduced to be cyanidin hexoside. Finally, compound **22** was confirmed as cyanidin-3-glucoside by comparison of the chromatographic and high-resolution MS data with those of the authentic standard. Thus, based on the fragmentation ion search of cyanidin, the typical fragmentation ions of cyanidin (m/z 287, 241, and 213), and the typical losses of glycosyls (C_6_H_10_O_5_ for hexosyl, C_6_H_10_O_4_ for deoxyhexosyl, and C_5_H_8_O_4_ for pentosyl), a series of cyanidin glycosides was identified. Similarly, other anthocyanins with aglycones of delphinidin and pelargonidin were identified with this strategy.

**Identification of other phenolic glycosides**. Other phenolic glycosides with various aglycones were similarly identified with this strategy.

**Identification of phenolic acids**. The typical fragmental pathway of the phenolic acids was the neutral loss of a CO_2_ (44 Da) in ESI negative ion mode. Herein, neutral loss of CO_2_ was detected in the MS/MS spectral data of caffeic acid and coumaric acid [24].

### 3.2. Quantitative Analysis of the Anthocyanins of R. meyeri Fruits

The content of the total anthocyanins was determined with a spectrophotometer and calculated using linear regression. The anthocyanins showed good linear relationships (R^2^ ≥ 0.9941) over the tested concentration range. The content of the methanol extract from the *R. meyeri* fruits was 34.37 ± 0.01 mg/g. The content of the anthocyanin fraction reached 137.65 ± 0.68 mg/g, and its content increased by nearly four times compared with the methanol extract. The result indicates that AB-8 macroporous resin can quickly and efficiently enrich and purify the anthocyanins of *R. meyeri* fruits.

The four main anthocyanins of the methanol extract and the anthocyanin fraction from *R. meyeri* fruits were quantified with UPLC-MS/MS. In the optimized chromatographic conditions, the four anthocyanins had the better resolution and peak shape, as shown in Figure 3.

The four anthocyanins show good linear relationships (*R*^2^ ≥ 0.9941) in the measured concentration ranges. The limit of quantification (LOQ) and limit of detection (LOD) of the four anthocyanins were calculated based on the signal-to-noise ratios of 10 and 3, respectively. Their LOQ and LOD ranged from 7.09 to 18.44 ng/mL and 2.36 to 5.38 ng/mL, respectively. The intra-day precision was calculated by measuring six replicates of the mixture solution containing four anthocyanins standard in a day. The inter-day precision was calculated by measuring the four anthocyanin standards for three consecutive days. The RSD values of the intra-day ranged from 1.09 to 4.65, and the RSD values of the inter-day were in the range of 1.17–5.57%. The six samples of *R. meyeri* fruits were prepared using the same method, and their repeatability was evaluated. The RSD values of repeatability ranged from 3.80% to 5.91%. The sample stability was determined for 0, 2, 4, 8, 12, and 24 h at room temperature, and the RSD values were in the range of 2.68–5.95%. The spiked recoveries were determined by adding the four anthocyanin mixed standards to a sample of known concentration. The recoveries were between 83% and 94%, and their RSD values ranged from 4.27 to 5.77%. The validation results show that the above UPLC-MS/MS method can determine the four anthocyanins efficiently, sensitively, and accurately, as shown in Table 4 and Table 5. Among the four anthocyanins, the content of cyanidin-3-*O*-rutinoside was the highest, at amounts of 22.01 ± 0.37 mg/g, while the content of delphindin-3-*O*-rutinoside was the lowest, at amounts of 0.06 mg/g. In a previous study, delphindin-3-*O*-rutinoside was the major anthocyanin in the black currant (*Ribes nigrum* L.) fruits, comprising 40.0–49.4% of the total anthocyanins [2,25]. Though the berries of the two *Ribes* plants are dark black fruits and are very similar, their anthocyanin contents differ significantly, which will impact their biological properties.

### 3.3. Antioxidant Activities

The antioxidative activities of *R. meyeri* fruits were investigated using ABTS and DPPH radical scavenging methods, as shown in Table 6. The result showed that the anthocyanin fraction of *R. meyeri* fruits had a higher DPPH radical scavenging rate than the methanol extract. The correlation coefficient between the anthocyanin content and the DPPH radical scavenging rate was 0.804. The results suggest that anthocyanins may be the active component in scavenging DPPH free radicals. For the ABTS radical, the methanol extract had a higher scavenging rate than the anthocyanin fraction, and the correlation between the anthocyanin content and the ABTS free radical scavenging rate was relatively low (correlation coefficients of 0.690). As previously reported, ascorbic acid, phenolic compounds, and anthocyanin are the most widespread antioxidants in berry fruits, particularly in *Ribes nigrum* [26,27]. For ABTS radicals, the methanol extract of *R. meyeri* fruits may have lost some antioxidant ingredients, such as ascorbic acid or other phenolics, during the enrichment of the anthocyanins with the AB-8 macroporous resin. In vitro analyses such as DPPH and ABTS radical scavenging are widely used to evaluate the antioxidant potentials of foods due to their speed and low cost. However, these in vitro methods have limitations. As previously indicated, in vitro tests for antioxidant capacity of different fruits showed poor correlation with in vivo radical scavenging capacity or physiological effects due to poor absorption and metabolism [28,29]. In future research, we will further investigate the in vivo antioxidant activity of phenolic compounds from *R. meyeri* fruits to clarify their mechanism of action.

### 3.4. α-Amylase, α-Glucosidase, and PTP1B Inhibitory Activities

The inhibitory activities of *R. meyeri* fruits on *α*-glucosidase, *α*-amylase, and PTP1B are shown in Table 6. For *α*-glucosidase, the methanol extract does not show inhibitory activity, but the anthocyanin fraction shows higher *α*-glucosidase inhibitory activity than acarbose. The correlation coefficient between the anthocyanin content and the *α*-glucosidase inhibitory activity is 0.874, which is probably due to the higher content of anthocyanin (in particular, cyanidin-3-*O*-glucoside and cyanidin-3-*O*-rutinoside) in the anthocyanin fraction. These results suggest that anthocyanins, especially cyanidin-glycoside, are efficient *α*-glucosidase inhibitors. The methanol extract and the anthocyanin fraction exhibit significant *α*-amylase inhibitory activities, and their activities are higher than acarbose. Although the anthocyanin content of the anthocyanin fraction is much higher than that of the methanol extract, their amylase inhibitory activities are very similar. The correlation coefficient between the anthocyanin content and the *α*-amylase inhibitory activity is 0.773. This may suggest that other phenolics of the *R. meyeri* fruit play a role in the amylase inhibitory activity. These results are similar to the previous study, which indicated that the anthocyanins in black currants regulate postprandial hyperglycemia primarily by inhibiting *α*-glucosidase, while other phenolics modulate salivary *α*-amylase, glucose uptake, and sugar transporters [6]. The previous study indicated that the excessive inhibition of pancreatic *α*-amylase could result in the abnormal bacterial fermentation of undigested carbohydrates in the colon, which could be responsible in part for the side effects of acarbose [30]. Therefore, combinations of *R. meyeri* fruit extract with low-dose acarbose could be employed as an alternative antidiabetic therapy, reducing the side effects of acarbose. Although there is a high correlation coefficient (0.810) between the anthocyanin content and the PTP1B inhibitory activity, the methanol extract and the anthocyanin fraction show lower PTP1B inhibitory activity than sodium vanadate. The inhibitory activity of phenolics, especially anthocyanins, against α-amylase and *α*-glucosidase has been reported to be highly related to their binding affinity to the enzymes. The structure of phenolic compounds, including aglycone structures, substitution groups, and sugar side chains, can affect their enzymatic activity [31,32,33]. The results of this study suggest that the phenolics of *R. meyeri* fruits exhibit antidiabetic effects through different targets. The anthocyanins had the better effect on hypoglycemic activity by inhibiting *α*-glucosidase, and some of the other phenolics may be the active components of *α*-amylase inhibition.

### 3.5. The Effect of R. meyeri Fruits Anthocyanin on Glucose Consumption of 3T3-L1 Adipocytes

The effect of *R. meyeri* fruits’ anthocyanin fractions on the glucose uptake of 3T3-L1 preadipocytes is shown in Figure 4. Compared with the control group, the anthocyanin fractions (12.5 and 25 μg/mL) promoted glucose uptake in 3T3-L1 adipocytes in the conditions without insulin. After adding insulin (100 nM), the anthocyanin fraction showed an antagonistic action on insulin and inhibited glucose consumption of the 3T3-L1 adipocytes. These results indicate that the anthocyanin fractions of *R. meyeri* fruits could significantly promote the glucose uptake of 3T3-L1 adipocytes.

## 4. Conclusions

In this study, 26 anthocyanins, 7 phenolic acids, and 9 flavonoids were identified from *R. meyeri* fruits using HPLC-QTOF-MS/MS. The contents of four anthocyanins were determined under optimized mass spectrometry conditions. Among the four anthocyanins, the content of cyanidin-3-*O*-rutinoside was the highest. After purification with AB-8 macroporous resin, the anthocyanin content in the anthocyanin fraction increased by approximately four times, indicating that AB-8 macroporous resin can quickly and efficiently purify and enrich the anthocyanins of *R. meyeri* fruits. In ABTS and DPPH radical scavenging assays, the activities of the methanol extract and anthocyanin fraction from *R. meyeri* fruits were lower than that of the ascorbic acid, and their PTP1B inhibitive effects were lower than that of sodium vanadate. However, they exhibited significant inhibitory activity on *α*-amylase and *α*-glucosidase. The anthocyanin fraction from *R. meyeri* fruits significantly promoted glucose consumption in 3T3-L1 adipocytes. *R. meyeri* fruits may play a hypoglycemic role in different targets, and the anthocyanin fraction has the better activity, which may be related to the high content of anthocyanin in them. Therefore, *R. meyeri* fruits are potential hypoglycemic small berries. It is necessary to elucidate the hypoglycemic mechanisms of the bioactive compounds from *R. meyeri* fruits, especially in in vivo studies using animal models.

## Figures and Tables

**Figure 1 foods-12-02406-f001:**
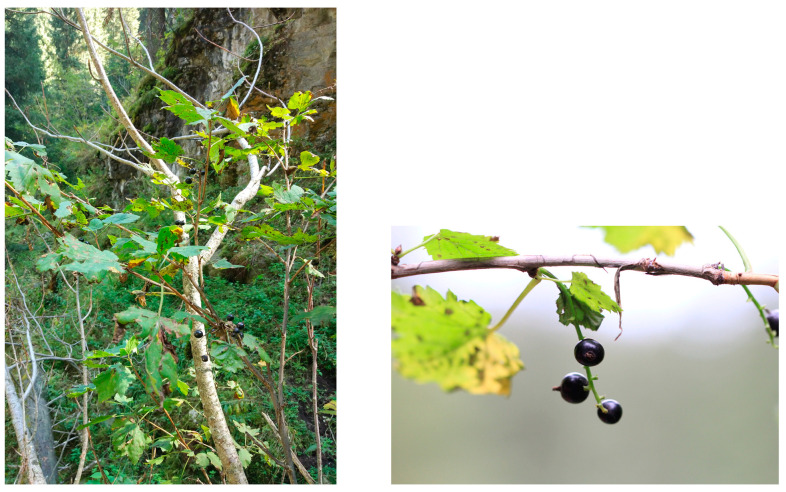
*Ribes meyeri* and fruits.

**Figure 2 foods-12-02406-f002:**
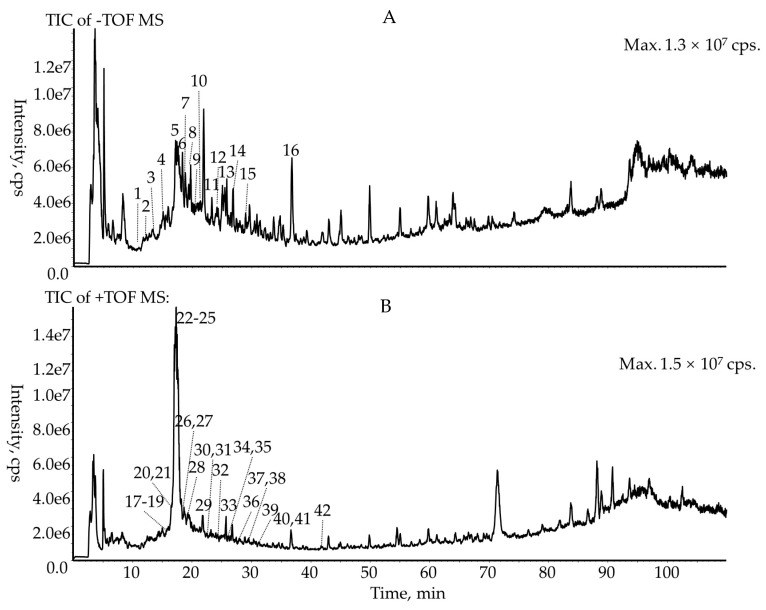
TIC chromatogram of *R. meyeri* fruits extract: (**A**) negative ion model, (**B**) positive ion model.

**Figure 3 foods-12-02406-f003:**
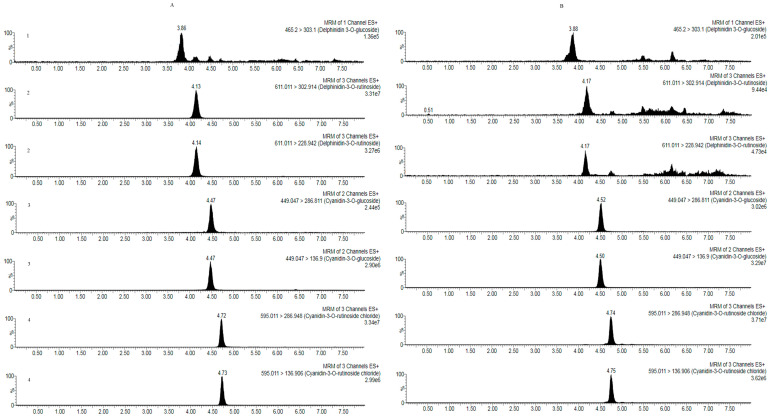
The MRM; chromatograms of (1) Delphinidin-3-*O*-glucoside; (2) Delphinidin-3-*O*-rutinoside; (3) Cyanidin-3-*O*-glucoside; (4) Cyanidin-3-*O*-rutinoside; (**A**) Standard; (**B**) The extract of *R. meyeri* fruits.

**Figure 4 foods-12-02406-f004:**
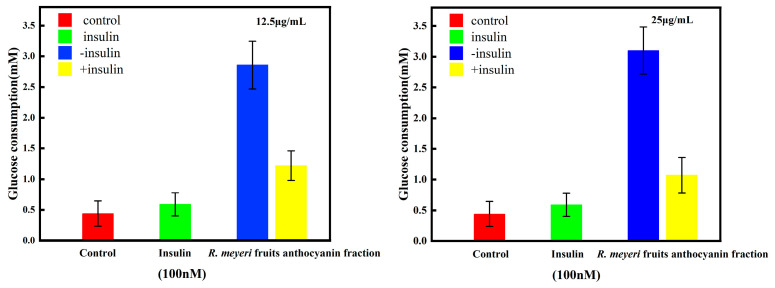
Effect of *R. meyeri* fruits’ anthocyanins on glucose consumption of 3T3-L1 preadipocytes.

**Table 1 foods-12-02406-t001:** List of quantitative compounds and MRM parameters.

Compound	Molecular Formula	Relative Molecular Mass	Ion Mode	Retention Time (min)	Parent Ion (m/z)	Daughter Ion (m/z)	Cone Hole Voltage (V)	Collision Energy (eV)
Delphinidin-3-*O*-glucoside	C_27_H_31_O_15_	465.1	ESI^+^	3.89	465.20	303.1 *	35	20
Delphinidin-3-*O*-rutinoside	C_27_H_31_O_16_	611.1	ESI^+^	4.13	611.01	302.9 *	58	34
						228.9	58	68
Cyanidin-3-*O*- glucoside	C_21_H_21_O_12_	449.1	ESI^+^	4.47	449.04	136.9 *	34	50
						286.8	34	66
Cyanidin-3-*O*- rutinoside	C_21_H_21_O_11_	595.0	ESI^+^	4.72	595.01	286.9 *	58	26
						136.9	58	70

* mean quantitative ions.

**Table 2 foods-12-02406-t002:** Identification of the phenolics of *R. meyeri* fruits.

No.	*t*_R_ (min)	Molecular Formula	[M-H]^−^	Major MS^2^ Ions	Error (ppm)	Identification
1	10.97	C_13_H_16_O_10_	331.0672	169, 125	3.70	Galloyl Hex
2	12.18	C_15_H_14_O_7_	305.0670	261, 219, 179, 167, 125	4.66	Gallocatechin/Epigallocatechin
3	13.18	C_13_H_16_O_8_	299.0775	239, 179, 137	4.27	Benzoyl Hex
4	15.47	C_15_H_18_O_8_	325.0933	163, 145, 119	4.63	Hydrocinnamoyl Hex
5	17.81	C_27_H_30_O_15_	593.1525	285, 284, 255, 227	4.05	Luteolin-Hex-DeHex
6	18.11	C_15_H_14_O_6_	289.0720	245, 205, 203, 109	4.62	Catechin
7	18.91	C_15_H_18_O_8_	325.0934	163, 145, 119	4.94	Hydrocinnamoyl Hex
8	19.96	C_15_H_18_O_8_	325.0934	163, 145, 119	4.94	Hydrocinnamoyl Hex
9	20.34	C_9_H_8_O_4_	179.0347	135	4.55	Caffeic acid
10	21.15	C_15_H_14_O_6_	289.0719	245, 205, 203, 109	4.62	Epicatechin
11	23.23	C_33_H_40_O_20_	755.2058	609, 447, 301, 300, 271, 255, 243, 179, 151	3.81	Quercetin-DeHex-Hex-DeHex
12	24.87	C_9_H_8_O_3_	163.0397	119	4.47	Coumaric acid
13	25.73	C_27_H_30_O_16_	609.1474	301, 300, 271, 255, 243, 229, 179, 151	3.90	Quercetin-3-*O*-Hex-DeHex
14	26.80	C_21_H_20_O_12_	463.0891	301, 300, 283, 271, 255	4.31	Quercetin-3-*O*-Hex
15	29.69	C_15_H_12_O_8_	319.0463	183, 153, 139	3.56	Dihydromyricetin
16	36.73	C_34_H_24_O_22_	301.0353	271, 255, 229, 179, 151	3.39	Quercetin

Hex: hexose; DeHex: deoxyhexose; Pen: pentose.

**Table 3 foods-12-02406-t003:** Identification of the anthocyanins of *R. meyeri* fruits.

No.	*t*_R_ (min)	Molecular Formula	[M+H]^−^	Major MS^2^ Ions	Error (ppm)	Identification
17	15.77	C_33_H_41_O_20_	757.2218	611, 449, 433, 287, 241, 213	4.26	Cyanidin-Hex-Hex-DeHex
18	15.81	C_33_H_41_O_20_	757.2222	611, 449, 433, 287, 241, 213	4.79	Cyanidin-Hex-Hex-DeHex
19	15.88	C_32_H_39_O_20_	743.2065	581, 449, 303, 287, 241, 213	4.82	Cyanidin-Hex-Pen-Hex
20	16.47	C_27_H_31_O_16_	611.1633	449, 287, 241, 213	4.32	Cyanidin-Hex-Hex
21	16.72	C_32_H_39_O_19_	727.2116	581, 449, 287, 241, 213	4.94	Cyanidin-Hex-Pen-DeHex
22	17.00	C_21_H_21_O_11_	449.1097	287, 241, 213	4.15	Cyanidin-3-glucoside
23	17.18	C_26_H_29_O_15_	581.1528	449, 287, 241, 213	4.65	Cyanidin-Hex-Pen
24	17.48	C_32_H_39_O_19_	727.2112	581, 449, 287, 241, 213	4.39	Cyanidin-Hex-Pen-DeHex
25	17.70	C_27_H_31_O_15_	595.1683	449, 287, 241, 213	4.29	Cyanidin-Hex-DeHex
26	18.81	C_21_H_21_O_10_	433.1148	271, 225, 197	4.33	Pelargonidin-Hex
27	18.83	C_33_H_41_O_20_	757.2220	611, 449, 433, 287, 241, 213	4.53	Cyanidin-Hex-Hex-DeHex
28	19.91	C_20_H_19_O_10_	419.0985	287, 241, 213	2.93	Cyanidin-Pen
29	22.03	C_21_H_21_O_10_	433.1145	287, 241, 213	3.64	Cyanidin-DeHex
30	23.13	C_32_H_39_O_20_	743.2069	611, 465, 449, 303, 257, 229	5.40	Cyanidin-Hex-Hex-Pen
31	23.16	C_33_H_41_O_20_	757.2223	611, 465, 303, 257, 229	4.92	Delphinidin-Hex-DeHex-DeHex
32	24.30	C_26_H_29_O_16_	597.1478	465, 303, 257, 229	4.67	Delphinidin-Hex-Pen
33	25.71	C_27_H_31_O_16_	611.1632	465, 303, 257, 229	4.15	Delphinidin-Hex-DeHex
34	26.30	C_21_H_21_O_12_	465.1045	303, 257, 229	3.76	Delphinidin-Hex
35	26.70	C_21_H_21_O_12_	465.1046	303, 257, 229	3.97	Delphinidin-Hex
36	27.69	C_15_H_11_O_5_	271.0613	253, 225, 197, 150	4.43	Pelargonidin isomer
37	29.33	C_15_H_11_O_5_	271.0608	253, 225, 197, 150	2.58	Pelargonidin isomer
38	29.70	C_15_H_11_O_5_	271.0611	253, 225, 197, 150	3.69	Pelargonidin isomer
39	31.30	C_15_H_11_O_5_	271.0611	253, 225, 197, 150	3.69	Pelargonidin isomer
40	36.54	C_15_H_11_O_7_	303.0511	257, 229, 201, 165, 153	3.86	Delphinidin
41	36.93	C_15_H_11_O_5_	271.0610	253, 225, 197, 150	3.32	Pelargonidin isomer
42	41.71	C_15_H_11_O_6_	287.0563	241, 213	4.48	Cyanidin

Hex: hexose; DeHex: deoxyhexose; Pen: pentose.

**Table 4 foods-12-02406-t004:** Linear regression equation, limit of quantification (LOQ), and limit of detection (LOD) for four anthocyanins.

Anthocyanins	Regression Equation	R^2^	LOQ (ng/mL)	LOD (ng/mL)	Linear Range (ng/mL)
Delphinidin-3-*O*-glucoside	*y* = 26.0*x* − 322.6	0.9941	16.98	3.01	10–1500
Cyanidin-3-*O*-glucoside	*y* = 312.6*x* − 3160.1	0.9980	7.09	2.36	10–1500
Delphinidin-3-*O*-rutinoside	*y* = 6260.1*x* − 24904.0	0.9969	13.86	4.55	10–1500
Cyanidin-3-*O*-rutinoside	*y* = 3216.0*x* − 8827.9	0.9957	18.44	5.38	10–1500

In the linear regression equation, *y* represents the peak area of quantitative ions, and *x* represents the concentration of anthocyanins.

**Table 5 foods-12-02406-t005:** The four anthocyanin contents of *R. meyeri* fruits (mg/g, *n* = 3).

Sample	Delphinidin-3-*O*-glucoside	Cyanidin-3-*O*-glucoside	Delphinidin-3-*O*-rutinoside	Cyanidin-3-*O*-rutinoside
*Ribes meyeri* fruits methanol extract	2.27 ± 0.05	4.76 ± 0.02	0.06 ± 0.00	22.01 ± 0.37
*Ribes meyeri* fruits anthocyanin fraction	5.99 ± 0.09	26.70 ± 0.59	0.09 ± 0.00	79.36 ± 1.01

**Table 6 foods-12-02406-t006:** Antioxidative and antidiabetic activities of *R. meyeri* fruits (Expressed as IC_50_, μg/mL, *n* = 3).

Sample		DPPH Radical Scavenging Rate	ABTS Radical Scavenging Rate	*α*-Glucosidase Inhibitory Activity	*α*-Amylase Inhibitory Activity	PTP1B Inhibitory Activity
*R. meyeri* fruits methanol extract		111.22 ± 2.28	4.06 ± 0.32	-	0.35 ± 0.01	744.60 ± 6.00
*R. meyeri* fruits anthocyanin fraction		74.22 ± 8.25	9.89 ± 0.87	56.67 ± 0.56	0.30 ± 0.01	218.72 ± 0.49
Positive control	Ascorbic acid	5.97 ± 0.06	2.61 ± 0.12			
	Acarbose			330.23 ± 1.03	3.91 ± 0.05	
	Sodium vanadate					1.46 ± 0.40

## Data Availability

Data are contained within the article.

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
