# Peer review of "Phenolic Profiles, Antioxidant, and Hypoglycemic Activities of Ribes meyeri Fruits"

_foods, 2023, doi:10.3390/foods12122406_

Round 1

Reviewer 1 Report

The manuscript can be improved, accepting the observations made in sections 2.7, 3.4 and 3.5

Author Response

Point 1: line 167, mention that the statistical analysis was applied to the results.

Response 1: We have added statistical analysis method. 

Point 2: line 272, antioxidant activities, possible explanation of these results.

Response 2: we have analyzed the experimental results and explained the possible causes.

Point 3: line 281, α-Amylase, α-glucosidase, and PTP1B inhibitory activities, possible explanation of these results.

Response 3: we have analyzed the experimental results and explained the possible causes.

Reviewer 2 Report

Line 104: "The anthocyanin content was determined according to the method [14] with some modifications." - Please, specify the name of the method or improve the sentence.

Figure 1 - It needs some improvements in the quality.

Table 6 needs some clarifications for a better understanding.

Author Response

Point 1: Line 104: "The anthocyanin content was determined according to the method [14] with some modifications." - Please, specify the name of the method or improve the sentence.

Response 1: We have added the name of the method- the pH-differential method and added the relevant information.

Point 2: Figure 1 - It needs some improvements in the quality.

Response 2: We have corrected as mentioned.

Point 3: Table 6 needs some clarifications for a better understanding.

Response 3: We have corrected as mentioned.

Reviewer 3 Report

In the manuscript ID: Foods-2398065, the authors analyzed phenolic components from Ribes mayeri fruits using HPLC-QTOFMS/MS and determined antioxidative activities, inhibitory activity against α-amylase and α-glucosidase, and glucose uptake of 3T3-L1 adipocytes of methanolic extract and crude anthocyanin fraction. The objective and approach in this study is appropriate, and new information on phenolic compounds from Ribes mayeri fruits is included. Each comment on this manuscript is described below.

1.     Figure 3 isn’t shown in this manuscript, though authors wrote it in page 5 line 181. Is it Figure 2? Figure 4 and 5 should be renumbered to Figure 3 and 4.

2.     Table 2, No. 1: 1254 is strange as major MS/MS ion.

3.     Table 3, No. 36, 37, 38, 39, and 41. Five peaks are identified as pelargonidin isomer. In general, the retention time of pelargonidin is longer than that of cyanidin, separated by ODS column. As authentic pelargonidin is commercially available, the peak of pelargonidin should be checked.

4.     Page 10, line 294: Phenolic acids and flavonoids are different from each other, and number of compounds should be counted independently.

Author Response

Point 1:  Figure 3 isn’t shown in this manuscript, though authors wrote it in page 5 line 181. Is it Figure 2? Figure 4 and 5 should be renumbered to Figure 3 and 4.

Response 1: Thanks. It's our carelessness. Two pictures of Ribes mayeri and their fruit are added in figure 1. All the figures are double-checked.

Point 2: Table 2, No. 1: 1254 is strange as major MS/MS ion.

Response 2: we have corrected as mentioned above, and changed 1254 to 125.

Point 3:  Table 3, No. 36, 37, 38, 39, and 41. Five peaks are identified as pelargonidin isomer. In general, the retention time of pelargonidin is longer than that of cyanidin, separated by ODS column. As authentic pelargonidin is commercially available, the peak of pelargonidin should be checked.

Response 3: Thank you very much. We carefully check the method section. The LC column is actually a "Prodigy Phenyl-3 column". We have corrected in the manuscript. See Line 136.

Point 4:   Page 10, line 294: Phenolic acids and flavonoids are different from each other, and number of compounds should be counted independently.

Response 4: we have revised as mentioned above.